# Genetic Polymorphism of *PTPN22* in Autoimmune Diseases: A Comprehensive Review

**DOI:** 10.3390/medicina58081034

**Published:** 2022-08-02

**Authors:** Kalthoum Tizaoui, Jae Il Shin, Gwang Hun Jeong, Jae Won Yang, Seoyeon Park, Ji Hong Kim, Soo Young Hwang, Se Jin Park, Ai Koyanagi, Lee Smith

**Affiliations:** 1Department of Basic Sciences, Division of Histology and Immunology, Faculty of Medicine Tunis, Tunis El Manar University, Tunis 2092, Tunisia; kalttizaoui@gmail.com; 2Department of Pediatrics, Yonsei University College of Medicine, Seoul 03722, Korea; shinji@yuhs.ac; 3College of Medicine, Gyeongsang National University, Jinju 52727, Korea; pearlmed15@gmail.com; 4Department of Nephrology, Yonsei University Wonju College of Medicine, Wonju 26426, Korea; kidney74@yonsei.ac.kr; 5Yonsei University College of Medicine, Seoul 06273, Korea; harryme1713@yonsei.ac.kr (S.P.); sooyoungsarah@yonsei.ac.kr (S.Y.H.); 6Department of Pediatrics, Eulji University School of Medicine, Daejeon 35233, Korea; fli018@hanmail.net; 7Research and Development Unit, Parc Sanitari Sant Joan de Déu, CIBERSAM, Dr. Antoni Pujadas, 42, Sant Boi de Llobregat, 08830 Barcelona, Spain; ai.koyanagi@sjd.es; 8ICREA, Pg. Lluis Companys 23, 08010 Barcelona, Spain; 9Centre for Health Performance and Wellbeing, Anglia Ruskin University, Cambridge CB1 1PT, UK; lee.smith@aru.ac.uk

**Keywords:** *PTPN22*, single nucleotide polymorphisms (SNPs), autoimmune diseases, genetic association, Lyp protein

## Abstract

It is known that the etiology and clinical outcomes of autoimmune diseases are associated with a combination of genetic and environmental factors. In the case of the genetic factor, the SNPs of the *PTPN22* gene have shown strong associations with several diseases. The recent exploding numbers of genetic studies have made it possible to find these associations rapidly, and a variety of autoimmune diseases were found to be associated with *PTPN22* polymorphisms. Proteins encoded by *PTPN22* play a key role in the adaptative and immune systems by regulating both T and B cells. Gene variants, particularly SNPs, have been shown to significantly disrupt several immune functions. In this review, we summarize the mechanism of how *PTPN22* and its genetic variants are involved in the pathophysiology of autoimmune diseases. In addition, we sum up the findings of studies reporting the genetic association of *PTPN22* with different types of diseases, including type 1 diabetes mellitus, systemic lupus erythematosus, juvenile idiopathic arthritis, and several other diseases. By understanding these findings comprehensively, we can explain the complex etiology of autoimmunity and help to determine the criteria of disease diagnosis and prognosis, as well as medication developments.

## 1. Introduction

An autoimmune disease refers to the condition of activating an abnormal immune response in our body, causing damage to the tissues or organs through continuous inflammation. The estimated prevalence of autoimmune diseases accounts for 4.5% of the general population and the number of new cases and mortality rates has increased over the past decades, which has increased the burden on society in spite of the development of immunosuppressants [1,2,3]. The phenotype of autoimmune diseases is heterogeneous, with over eighty autoimmune diseases such as rheumatoid arthritis, Grave’s disease, Hashimoto’s thyroiditis, Sjogren’s syndrome, and less common diseases identified [4]. The cause of autoimmune diseases is not well understood, and physicians have suggested that both environmental and genetic factors, as well as other factors such as infection, are accountable for the diseases. Recent advancements in the field of genetic epidemiology and genome-wide association (GWA) studies have made it possible to discover the genetic variants and genes associated with the diseases [5]. Since most autoimmune disorders present similar clinical features, some of the common genes are known to be strongly correlated with the autoimmunity [5,6]. Investigating the characteristics of these common genes can give clues in identifying the diseases with unknown etiology.

The protein tyrosine phosphatase non-receptor 22 gene (*PTPN22*) is one of the candidate susceptibility genes for autoimmune diseases. It is located on chromosome 1p13.3-13.1 and encodes the protein called lysine tyrosine phosphatase (Lyp) [7,8]. The single nucleotide polymorphism (SNP) *PTPN22* C1858T (rs2476601) in exon 14 is mainly associated with the onset of autoimmune diseases. A change in cytosine to thymidine at nucleotide 1858 resulted in a change in amino acids from arginine to tryptophan at codon 620 (R620W), and a change in the Lyp protein interrupts the cell signaling by disrupting the function of the T cell antigen receptor, mostly found in various types of lymphoid tissues. The Lyp protein is important in the prevention of spontaneous T cell activation, development, and inactivating of T-cell-receptor-associated kinases and their substrates [9]. Since Botinni et al. first reported the association between *PTPN22* gene variants and type 1 diabetes mellitus (T1DM), studies on other diseases such as rheumatoid arthritis (RA), systemic lupus erythematosus (SLE), autoimmune thyroid diseases, and vitiligo have been published successively. This strongly reflects the association of *PTPN22* SNPs with autoimmunity.

Previously, our team systematically analyzed the association between the *PTPN22* polymorphism and autoimmune diseases using the Bayesian approach, then reviewed the immunologic functions of the *PTPN22* polymorphism [6,10]. To broaden the perspective of this association, in this review, we aim to summarize how the *PTPN22* gene and its variants are associated with the onset and progress of a large set of autoimmune diseases (Table 1).

## 2. *PTPN22* C1858T Associations with Autoimmune Diseases

### 2.1. Association of PTPN22 C1858T with Type 1 Diabetes

Bottini et al. first confirmed the association between *PTPN22* R620W polymorphism and T1DM [8]. Afterward, Heneberg et al. confirmed that the 1858T allele serves as a risk allele for latent autoimmune diabetes in adults (LADA) [11]. They also confirmed gender-related differences in the frequency of some *PTPN22* polymorphisms (but not c.1858C>T) in LADA. Re-analysis of the genetic association between the R620W variant and the risk of T1DM under Bayesian approaches false-positive report probability (FPRP) or Bayesian false discovery probability (BFDP) come in support of these findings. Out of 22 comparisons from observational studies, 19 (86.4%) comparisons had noteworthy findings [6].

At the cellular level, in addition to its impact on T cells, the *PTPN22* variant conferred a risk for T1DM by influencing B cell activation. The Lyp R620W variant increases the number of autoreactive B cells, promoting the onset of autoimmune pathologies through the internalization and the presentation of autoantigens to T lymphocytes [12,13]. The group of Habib reported that the presence of the Lyp R620W variant has an effect on the peripheral B cell homeostasis in heterozygous healthy controls, promoting a specific expansion of the transitional and anergic IgD+, IgM−, CD27−, and B cell populations [14]. They also reported reduced B cell receptor signaling and resistance to apoptosis in both the transitional and naive B cell compartments in T1DM patients, irrespective of the presence of the *PTPN22* genotype [14]. *PTPN22* C1858T influences innate and adaptive immunity by perturbing the homeostasis of B cells and Toll-like receptor (TLR)-9-mediated response in T1DM patients [15]. In addition, the Lyp variant may influence cytokine production [16]. Meta-analysis investigations showed that in the Caucasian population, T cells of some patients with T1DM are characterized by a defect in IL-2 production [17,18,19,20].

Interestingly, *PTPN22* acts in T1DM through the modulation of Treg cells. Both in vivo and in vitro experiments using *PTPN22* knock-out mice showed that *PTPN22* plays a key role in Treg induction and acts mainly through modulating the threshold of the T cell activation [21]. Unexpectedly, some experiments on animal models suggested a protective role of *PTPN22* in T1DM. Overexpression of *PTPN22* resulted in attenuated Th1 differentiation at low strength T cell receptor (TCR) stimulation and protected mice from a model of diabetes [22]. NOD mice where *PTPN22* expression was targeted by a knock-down genetic approach were protected from autoimmune diabetes. Surprisingly, Yeh et al. found that *PTPN22* transgenic NOD mice that overexpressed *PTPN22* were also protected from T1DM [22]. Experiments by Lin et al. confirmed previous findings showing that in contrast to *PTPN22* knocked-down mice, *PTPN22* R619W NOD mice showed accelerated T1DM and increased prevalence and elevated titer of insulin [23]. Thus, either downregulation or overexpression of *PTPN22* had a protective effect from T1DM in NOD mice. *PTPN22* knock-down in NOD mice resulted in T1DM prevention possibly because of a dominant effect of *PTPN22* on the Treg cells [24]. As it was shown in several mouse models of diverse genetic backgrounds, the number and functionality of Treg cells increase when *PTPN22* levels reduce [24,25,26]. Studies in humans found that the *PTPN22* variant conferred significant risk to T1DM; however, one meta-analysis showed a protective effect [27]. Re-analysis of previous results by using Bayesian approaches did not confirm this exception, thus this meta-analysis may meet one of several meta-analysis limitations [6]. Overall, results suggested a significant risk conferred by the *PTPN22* 620W variant in T1DM.

### 2.2. Association of PTPN22 C1858T with Rheumatoid Arthritis

*PTPN22* is the strongest non-HLA genetic predisposition factor in RA. The first report on the significant association between the *PTPN22* 1858T allele and RA was published by Begovich and co-workers in 2004 [28]. The homozygous *PTPN22* 1858C variant is shown to increase the risk of RA by twice that of the 1858T variant, from which it can be interpreted that this variant is a co-dominant allele [28,29,30,31,32]. The data in RA show a dosage effect of the *PTPN22* risk allele [33]. Several studies focused on the association of the *PTPN22* variant with RA risk and its clinical features. The *PTPN22*R620W allele is associated with seropositive diseases [30,33], anti-citrullinated protein antibodies (ACPA) [34,35], erosive diseases [36], and earlier disease onset [37]. In a stratified meta-analysis, *PTPN22*C1858T was more common in RF-positive than in RF-negative patients and was also more common in patients with anti-CCP antibodies than those without [38]. Although *PTPN22* 1858T is associated with both autoantibody seropositive and seronegative RA, most studies have reported stronger associations of *PTPN22* with RF-positive or ACPA-positive RA [28,35,37]. A GWAS confirmed that *PTPN22* 1858T is only of genome-wide significance in ACPA-positive RA patients [39]. Although some studies have detected an effect of *PTPN22* on the presence of radiographic erosions or the rate of joint destruction in RA, a meta-analysis indicated no such association in either anti-CCP antibody seropositive or seronegative individuals [36,40,41]. Most studies showed an earlier (2–7.5 years) age at onset of RA in carriers of the *PTPN22* 1858T allele, but not all studies showed the same effect [36,42,43]. Several limitations related to experiment design and methods raise discrepancies between results. Re-analysis of previous findings using Bayesian approaches showed that 32 (82.1%) of the 39 comparisons from observational studies and one meta-analysis of GWAS had noteworthy findings by FPRP or BFDP [6].

At the molecular level, genetic polymorphism in *PTPN22* may contribute to RA disease through a number of distinct mechanisms. The deficiency of *PTPN22* function could contribute to the chronic activation of antigen-specific, class-II-restricted CD4+ T cells and other types of effector T cells which contribute to driving the inflammatory process within the synovium [44,45]. In a mouse model, *PTPN22* could regulate vimentin-dectin-1 driven uptake and presentation of autoantigens, in addition to cytokine secretion [44]. Serum autoantibodies against citrullinated vimentin, common in RA patients, have been shown to promote osteoclastogenesis and bone resorption [44]. Human cells expressing *PTPN22* Trp620 have deficient TLR-induced IFN production, and *PTPN22* dysfunction results in lowering thresholds for TCR signaling [46,47]. In a model of IL-1β-dependent synovial inflammation, overexpression of transgenic human *PTPN22* Trp620 in mice impaired amelioration of inflammatory arthritis by treatment with an IFN-inducing TLR agonist [47]. Autoimmune pathogenesis promoted by *PTPN22* 1858T probably involves concerted anomalies in the differentiation of T cell subsets, B cell repertoire, and the balance between immunoregulatory and proinflammatory cytokine production.

In addition to the C1858T polymorphism, *PTPN22* variants have been found in RA association, particularly in populations with low frequencies of the 1858T allele. a meta-analysis reports that the *PTPN22* gene C1858T (rs2476601) SNP increases RA risk, especially in Caucasians and Africans [48].

### 2.3. Association of PTPN22 C1858T with Juvenile Idiopathic Arthritis

Juvenile idiopathic arthritis, the common type of autoimmune arthritis in children under 16, is also known to be associated with the *PTPN22* 1858T allele [49]. Several meta-analyses and SNP replication studies proved the significant contributions of *PTPN22* 1858T to the risk of JIA onset [50,51,52,53]. The *PTPN22* 1858T conferred risk for oligoarticular and RF-negative polyarticular JIA in white European, American, and Australian individuals [50,52]. A meta-analysis by Kaalla et al. reported that the 1858T allele was associated with RF-positive polyarticular JIA, but not with systemic-onset or enthesitis-related JIA [53]. Re-analysis of previous results including five studies with 15 genotype and allele comparisons showed that 9 (60%) and 1 comparison from a GWAS meta-analysis had noteworthy findings by FPRP or BFDP Bayesian approaches [6].

### 2.4. Association of PTPN22 C1858T with Systemic Lupus Erythematosus

In 2004, Kyogoku and colleagues first reported that the *PTPN22* 1858T allele is associated with SLE [54]. GWAS found an association of *PTPN22* 1858T with seropositive SLE in a case-only analysis and another study found a positive association with anti-cardiolipin IgG and a trend towards an increased frequency of *PTPN22* 1858T in patients with lupus nephritis or in individuals seropositive for anti-dsDNA autoantibodies [55,56]. In the case of SLE, both immune complex deposition and the direct effects of antibodies can contribute to this disease. Re-analysis of previous associations including seven observational studies with 15 genotypes and allelic comparisons reported that 13 (86.7%) of the 15 comparisons had noteworthy findings by FPRP or BFDP [6].

SLE is a systemic inflammatory disorder characterized by the production of autoantibodies, immune complex formation, and immune complex deposition in end-organs. The *PTPN22* 1858T allele has been demonstrated to be associated with lower IFN-γ and higher IFN-α levels in SLE [57]. As a consequence of dysregulated IFN-γ expression in SLE, patients carrying the 1858T risk variant may have enhanced IFN-α-mediated JAK-STAT signaling [58]. Pep and IFN-γ might cooperate to give rise to dysfunctional hematopoiesis. Animal models showed that the *PTPN22*W* polymorphism may also influence TCR signaling, augmenting the mediators implicated in the early events of the TCR-initiated response such as protein tyrosine phosphorylation and calcium mobilization [59]. It has been shown previously that TCR signaling was increased in SLE upon anti-CD3 monoclonal antibody (mAb) stimulation [60]. In addition, high *PTPN22* transcript numbers in CD8+ T cells correlated with poor prognosis of SLE and AAV [61].

### 2.5. Association of PTPN22 C1858T with Vasculitides

The *PTPN22* polymorphism is positively associated with microscopic polyangiitis (MPA) and granulomatosis with polyangiitis (GPA), formerly known as Wegener’s granulomatosis, but has not been reported in eosinophilic granulomatosis with polyangiitis (eGPA), formerly known as Churg–Strauss syndrome [62,63]. The association with GPA is stronger in patients with organ pathology (lung, kidney, eye, or peripheral nervous system) [63]. Several studies have documented and replicated a significant association of the 1858T allele with biopsy-proven giant cell arteritis (GCA) [64]. Intriguingly, two studies reported that *PTPN22* 1858T can protect against Behçet’s disease (BD) [65]. In Bayesian re-analysis, a total of four studies with 11 genotypic and allelic comparisons were included for ANCA-associated vasculitis. Out of 11 comparisons, 6 (54.5%) had noteworthy findings by FPRP or BFDP [6]. For the studies including subjects with GCA, re-analysis of observational studies and GWAS by Bayesian approaches revealed that among three comparisons, two were noteworthy [6].

### 2.6. PTPN22 C1858T in Autoimmune Thyroid Disease

In addition to T1DM, RA, SLE, JIA, and vasculitis, other autoimmune disorders such as autoimmune thyroid diseases (AITD), including Grave’s disease and Hashimoto’s disease, Addison’s disease, autoimmune thrombocytopenia, inflammatory bowel disease, vitiligo, etc., had a significant correlation with the *PTPN22* 1858T allele [38,66]. A meta-analysis showed that *PTPN22* C1858T is associated with the risk of Grave’s disease and Hashimoto’s thyroiditis in the overall study population. In addition, this polymorphism is associated with elevated AITD risk in Caucasians, but not in Asians [67]. A total of 212 Korean AITD patients were studied; interestingly, a minor allele of an SNP (rs12730735) and a haplotype (GGCTT) showed significant association with the susceptibility of AITD, especially with that of Hashimoto’s thyroiditis [68]. In Chinese AITD patients, Gong et al. reported rare missense mutation in *PTPN22* (NM_015967.5; c.77A > G; p.Asn26Ser) using whole-exome sequencing in Hashimoto’s thyroiditis, but *PTPN22* C1858T mutation was not confirmed [69].

### 2.7. PTPN22 C1858T in Autoimmune Skin Diseases

A study showed not only the significant association of the *PTPN22* C1858T with patients with psoriasis arthritis (PsA) (Odds ratio, 1.49; 95% confidence interval, 1.10–2.02) but also showed a greater number of deformed joints [70]. While most susceptibility loci identified in psoriasis (PsO) tend to be equally associated with skin psoriasis and with PsA, the 1858T allele *PTPN22* is weakly associated with general skin psoriasis whereas its association with PsA is statistically highly significant [71]. This suggests that *PTPN22* may influence more cells and pathways influenced in PsA, which has additional components in its pathogenesis compared to skin-restricted. Bowes et al. suggested that the differential association of *PTPN22* Trp620 with PsA vs. PsO depends on alterations in the function of CD8 T cells, which have been known to be influenced by *PTPN22* [71,72]. The known role of *PTPN22* in CD8 memory T cell function and IL-17-producing Th17 cell differentiation suggests the possibility that *PTPN22*-W620 contributes to differential phenotypes of Th17 in PsA vs. PsO or AS [73,74].

The T allele of the single nucleoid polymorphism (SNP) rs2476601 in the *PTPN22* gene is a risk factor for developing alopecia areata. However, more robust studies defining the ethnic background of the population of origin are required, so that the risk identified in the present study can be validated [75]. The *PTPN22* 1858T allele of SNP rs2476601 is also reported to be associated with an increased risk of generalized vitiligo [76,77,78]

### 2.8. PTPN22 C1858T in Other Autoimmune Conditions

Intriguingly, the allele was protective against two autoimmune disorders, Crohn’s disease (CD) and Behçet’s disease (BD) [79]. However, these reports did not contain large sample sizes, and in some cases, these associations have failed to replicate. In addition, there were no noteworthy findings by FPRP or BFDP in one study (two comparisons) of BD and one study (three comparisons) of AITD [6]. No association was observed between rs2476601 and autoimmune diseases of the liver and the bile duct, such as autoimmune hepatitis (AIH), primary biliary cholangitis (PBC), and primary sclerosing cholangitis, but further investigation is needed as not many studies were conducted [49,80,81].

Unlike in RA studies, the association with systemic sclerosis (SSc) is not affected by the presence of autoantibodies, as meta-analysis did not reveal a difference in allele frequency when comparing anti-centromere antibody seropositive and seronegative or anti-topoisomerase I autoantibody seropositive and seronegative SSc [38,82]. Bayesian approaches including two studies with three allelic comparisons analyzed the genetic impact of psoriasis (PsO) and did not verify noteworthiness by means of both FPRP and BFDP estimations [6]. Three studies including patients with SSc analyzed seven genotypic and allelic comparisons and did not show noteworthiness in terms of FPRP and BFDP estimations. Findings from patients with SS and AS included two studies, only one study for each, and did not verify noteworthiness by means of FPRP and BFDP estimations. However, a GWAS meta-analysis including one comparison showed noteworthy results for PsA [6]. It should be noted that meta-analyses from PsO, SS, SSc, and AS patients had several limitations related to the number and population size of included studies. Therefore, meta-analyses with larger sizes including different ethnicities and clinical features would give significant results.

A single case-control study that reported an association of *PTPN22* C1858T with idiopathic inflammatory myopathy in white individuals suggested that after stratification analysis the association was restricted to polymyositis and juvenile dermatomyositis, and not to dermatomyositis or myositis overlapping with another connective tissue disease [83]. In addition, *PTPN22* 1858T was not associated with dermatomyositis in a GWAS of patients with adult or juvenile dermatomyositis [84]. In the case of GD and MG, it is widely recognized that the disease-associated autoantibodies are pathogenic. A direct role for autoantibodies is less clear for Hashimoto’s thyroiditis, vitiligo, and rheumatoid arthritis, although recent studies suggest that the anti-citrulline antibodies may contribute directly to joint inflammation [85]. Interestingly, many of these *PTPN22*-associated diseases also appear to cluster together in families, suggesting that the *PTPN22* association reflects the involvement of common pathways in these disorders [86,87]. The C1858T polymorphism could contribute to the development of GD and HT in children, with a strong indication that females are pre-disposed to developing the disease and the T allele is the main risk factor [88]. Re-analysis of previous results by Bayesian approaches reported that among three studies from subjects with MG reporting five allelic comparisons, four (80%) of the five comparisons had noteworthy findings by FPRP or BFDP. Out of three studies with five comparisons included from patients with vitiligo, four (80%) of the five comparisons had noteworthy findings by FPRP or BFDP. For Addison’s disease, out of the three comparisons, two were noteworthy in terms of BFDP. For patients with endometriosis, one study with three co-dominant comparisons did not verify noteworthiness, except for one finding which was noteworthy by using BFDP. There were no noteworthy findings by FPRP or BFDP in one comparison of alopecia areata [6]. Re-analysis from patients with CD including five studies revealed that two (40%) of the five comparisons had noteworthy findings by FPRP or BFDP [6].

At the functional level, several studies tried to explain how *PTPN22* contributes to CD. *PTPN22* regulates intracellular signaling events and is induced by IFN-γ in human monocytes [89]. Knock-down of *PTPN22* alters the activation of inflammatory signal transducers, increasing the secretion of Th17-related inflammatory mediators [90]. This might explain on a functional level how the reduced *PTPN22* expression found in CD patients contributes to CD pathology. Spalinger et al. showed that TNFα levels are elevated in CD patients, decreasing *PTPN22* expression significantly; thus, TNFα is likely to play an even more important role in CD pathogenesis than IFN-γ [90]. In concordance with these findings, the C1858T polymorphism, which causes a gain of function, is protective in CD and attenuates the expression of proinflammatory cytokines [91,92]. *PTPN22* also is involved in the regulation of Src kinase and negatively controls the p38-MAPK/IL-6 pathway [93]. p38-MAPK activation and IL-6 secretion by antigen-presenting cells (APC) play a crucial role in the differentiation of CD4+ naive T cells into Th17 cells that are more and more regarded as the driving force of CD [94].

## 3. Other Polymorphisms in the *PTPN22* Gene Are Associated with Autoimmune Diseases

Not only the C1858T polymorphism but other *PTPN22* gene variants also have been investigated to be associated with autoimmune disorders. These associations especially are more prominent in specific populations with low frequency of the C1858T polymorphism and they are associated with resistance to certain autoimmune diseases, indicating the complexity of most autoimmune diseases [95]. In the Asian population, a systemic search for SNPs allowed identifying five SNPs in the *PTPN22* gene, while C1858T was not found (vide supra). Among these, two SNPs, G1123C and C2740T, showed allele frequencies of more than 5% [96].

The G1123C polymorphism (rs2488457) is located in the 5′ promoter region of *PTPN22,* and its function has not yet been characterized. The impact of this non-coding SNP on the transcription, stability, or translation of the mRNA remains to be fully clarified. It has been found that the SNP is associated with RA, JIA, the onset of acute T1D in Japanese and Korean subjects, latent autoimmune diabetes in Chinese patients, and UC [96,97,98,99,100]. Interestingly, rs2488457 was recently reported as a potential cis-expression quantitative trait loci (eQTLs) in whole blood from Spanish RA patients, and another study demonstrated that *PTPN22* expression is significantly decreased in whole blood from RA patients carrying the risk alleles of SNPs C1858T and G1123C compared to healthy controls [101,102].

The second (rs33996649) is a rare missense G788A mutation that does not co-occur with C1858T and encodes an R263Q substitution in the catalytic domain of the protein [103]. *PTPN22* G788A encodes a loss-of-function Arg263Gln substitution that changes the conformation of the active site and results in the reduced catalytic activity of *PTPN22* [103]. Therefore, the 788A allele displays a pattern of autoimmune disease association that is distinct from the 1858T allele in European populations. Single studies have so far shown no associations with SSc, GCA, IgA vasculitis, uveitis, or GD [64,104]. In contrast to 1858T, the 788A allele protects against both SLE and RA [104]. The 788A allele reduced risk for UC, which 1858T does not associate with, and the 788A does not associate with CD, with which the 1858T is protective against [92,104].

Recently, Gong et al., by using whole-exome sequencing in a Chinese Hashimoto’s thyroiditis pedigree, identified an extremely rare missense mutation in *PTPN22* (NM_015967.5; c. 77A > G; p. Asn26Ser) [69]. The missense mutation *PTPN22* (N26S) is located in the classical catalytic domain of the N-terminal protein tyrosine phosphatase. Little is known regarding its specific function; however, co-segregation analysis confirmed that all patients in this family were female, and authors linked this variant to Hashimoto’s thyroiditis [69]. Considering the female predominance in most of the autoimmune disorders associated with the *PTPN22* Trp620 variant, Nielsen et al. investigated the existence of cis-acting or sex-specific trans-acting factor/s (e.g., sex hormones) affecting the allele-specific expression of the *PTPN22* Arg620Trp polymorphism [105]. They report no effect of sex or pregnancy status on the relative expression of the *PTPN22* 1858T allele, indicating the absence of sex-specific trans-acting factor/s (e.g., sex hormones) [105].

Many SNPs were studied to assess the association between ethnicity and susceptibility to different autoimmune diseases. A Japanese study identified nine SNPs in the *PTPN22* gene and found minor alleles at rs1217412, rs1217388, rs1217407, and rs2488458 less frequent in autoimmune hepatitis patients compared with controls [80]. This is in contrast with a genome-wide association study where *PTPN22* was not related to autoimmune hepatitis patients of European descent [106]. In a study based on the Chinese Han population, rs1217414 and rs3811021 showed a strong association with both SLE and RA, while rs3765598 had a significant association with SLE only [107].

**Table 1 medicina-58-01034-t001:** Summary of the *PTPN22* C1858T polymorphism with different types of autoimmune diseases.

Type of Diseases	Summary of Immunologic Functions	References
T1DM	*PTPN22* variant produces Lyp R620W protein which increases the number of autoreactive B cells, promoting the autoimmune reactions by internalizing and presenting autoantigens to T lymphocytes.	[12,13]
	In T1DM patients with *PTPN22* variants, expansion of B cell along with reduced B cell receptor signaling and resistance to apoptosis were found.	[14]
	*PTPN22* 1858T variant disturbs the homeostasis of B cells and Toll-like receptor 9 mediated response, and also the cytokine production.	[15]
	*PTPN22* variant increases the number of Treg cells then modulates the threshold of T cell activation.	[21]
	Overexpression of *PTPN22* encodes Pep-decreased TCR-mediated effector cell responses then prevents the disease process.	[22]
	On the contrary, *PTPN22* knocked-down mice showed an acceleration of T1DM by elevating the titer of insulin.	[23]
RA	Lack of *PTPN22* function activates the antigen-specific, class-II-restricted CD4+ T cells and effector T cells, contributing inflammatory process in the synovium, accelerating the RA progress.	[44,45]
	Increased number of antibodies also promotes osteoclastogenesis and bone resorption.	[44]
	*PTPN22* Trp620-expressing human cells have lack of production of TLR-induced IFN production, have lowered threshold for TCR signaling.	[46,47]
SLE	In the C1858T polymorphism of *PTPN22*, the risk of SLE increased by lowering IFN-gamma rate and higher serum IFN-α activity.	[57]
	The 1858T variant may enhance IFN-α-mediated JAK-STAT signaling, and the increasing number of Pep and IFN-α results in dysfunctional hematopoiesis.	[58]
	*PTPN22*W* polymorphism influences TCR signaling and augments the TCR-initiated response to promote autoimmunity.	[59]
	Patients with SLE showed abnormality in TCR/CD3 monoclonal antibody stimulation.	[60]

## 4. Conclusions

To sum up, we can conclude that the SNPs of *PTPN22* play an important role in the onset of autoimmune diseases including T1DM, RA, JIA, PsA, SLE, SSc, AITD, and different forms of vasculitis. Not only the most well-known polymorphism of *PTPN22* at position 1858 called the *PTPN22* C1858T SNP has significant variant characteristics, but also new variants in a variety of genes, such as cytotoxic T-lymphocyte-associated protein 4 (CTLA4), tumor necrosis factor (TNF), interferon regulatory factor 5 (IRF5), etc., are recently reported to be associated with autoimmune disorders (Table 1). Recent large-scale GWAS studies support the strength of genetic factors on autoimmunity; nevertheless, this still needs to be supported with additional research.

## Data Availability

Not applicable.

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
