# Peer review of "Genetic Polymorphism of PTPN22 in Autoimmune Diseases: A Comprehensive Review"

_medicina, 2022, doi:10.3390/medicina58081034_

Round 1

Reviewer 1 Report

Title: Genetic Polymorphism of PTPN22 in Autoimmune Diseases: A Comprehensive Review

The manuscript is clear, relevant for the field and presented in a well-structured manner. The manuscript is scientifically sound

General

Language editing is need

Thera re multiple grammar errors

The main topic is about autoimmune diseases as a whole.

However, the author repeated the word “autoimmune disease “multiple times in single

Introduction

It is better to add a paragraph that describes what the authors meant by autoimmune diseases and what is the spectrum they cover as the word “autoimmune “is wide and includes many many disorders.

Line 75

It is not appropriate to start with diabetes as an autoimmune disease

85-120

A too long paragraph

Need to be summarized and divided into multiple paragraphs

The manuscript is need of some images to illustrate the text

It will be better to add few sentences about each disease discussed before the authors proceed to the PTPN22 role in the pathogenesis

213

The order of the diseases should be changed. where autoimmune rheumatic diseases first then autoimmune endocrine diseases …etc.

421

The authors mentioned SSc under the title of autoimmune skin disease. However, it is a systemic disease not limited to the skin

261

Autoimmune diseases is a term that differs from the term autoimmune diseases ...overlap may be present

So, this sentence is not appropriate “was protective against two autoinflammatory disorders “

Author Response

Thank you for the meticulous review. Here is the point-by-point response to your comments. We have tried our best to apply your comments.

Language editing is need Thera re multiple grammar errors

The main topic is about autoimmune diseases as a whole.

However, the author repeated the word “autoimmune disease “multiple times in single

Autoimmune diseases is a term that differs from the term autoimmune diseases ...overlap may be present

So, this sentence is not appropriate “was protective against two autoinflammatory disorders “

- We have performed a grammar check thoroughly.

- The term "autoimmune disease" was revised as "diseases."

- We edited it to "autoimmune diseases."

It is better to add a paragraph that describes what the authors meant by autoimmune diseases and what is the spectrum they cover as the word “autoimmune “is wide and includes many many disorders.

- We have expanded the description of autoimmune diseases in the introduction.

It is not appropriate to start with diabetes as an autoimmune disease

The order of the diseases should be changed. where autoimmune rheumatic diseases first then autoimmune endocrine diseases …etc.

- Although writing in the order of autoimmune rheumatic diseases and autoimmune endocrine diseases is reasonable, we have discussed Type 1 DM first because PTPN22 gene is known for its association with Type 1 DM and RA the most. 

A too long paragraph

Need to be summarized and divided into multiple paragraphs

- We have divided it into two paragraphs.

The manuscript is need of some images to illustrate the text

- We were unable to provide illustrations for several reasons and focused on the manuscript more. 

It will be better to add few sentences about each disease discussed before the authors proceed to the PTPN22 role in the pathogenesis

- We improvised the description of autoimmune diseases further. 

The authors mentioned SSc under the title of autoimmune skin disease. However, it is a systemic disease not limited to the skin

- We located systemic sclerosis to "other autoimmune disorders." 

Reviewer 2 Report

This manuscript also has valuable information but I think it is not comprehensive enough. it is mainly written about C1858T  and not all autoimmune disorders are covered. for example, other diseases like vitiligo, Alopecia areata, psoriasis, IBD, Behcet’s diseases, and Autoimmune hepatitis (AIH), ... are not covered completely. The author can check the below references to find out the other related disease 

https://www.intechopen.com/chapters/70958

https://www.nature.com/articles/srep29770

Indeed the SNPs are named non consistently, in some places they are named according to the nucleotide variation, some places according to the rs code, and some places according to the amino acid change, which causes misunderstanding. all the discussed SNPs should be named completely for the first time (example: rs2476601 (C1858T, R620W) and then just use one of the names like the rs code for all the remaining text for all the SNPs. 

one more thing is that there are other SNPs in PTPN22 gene which are not considered in this paper: rs1217412, rs1217388, rs1217407, and rs2488458, rs3811021, rs3789605 

 In introduction line 60, please add general information about all the pathogenic SNP in PTPN22. How many SNPs are discovered now, how many are coding or non-coding?

You should explain the PTPN22 gene structure and function in a separate heading in the paper 

Author Response

Thank you for your comments and the reference. We have done our best to apply all comments. Here is the point-by-point response. 

it is mainly written about C1858T  and not all autoimmune disorders are covered. for example, other diseases like vitiligo, Alopecia areata, psoriasis, IBD, Behcet’s diseases, and Autoimmune hepatitis (AIH), ... are not covered completely.

  • Psoriasis, alopecia areata, Crohn's disease, and Behcet's disease, are discussed in 2.7. and 2.8.
  • Vitiligo was added on 2.7. and autoimmune hepatitis was added on 2.8. 

Indeed the SNPs are named non consistently, in some places they are named according to the nucleotide variation, some places according to the rs code, and some places according to the amino acid change, which causes misunderstanding. all the discussed SNPs should be named completely for the first time (example: rs2476601 (C1858TR620W) and then just use one of the names like the rs code for all the remaining text for all the SNPs. 

  • We edited line 62 in the introduction mentioning the full name "PTPN22 C1858T (rs2476601)." Other SNPs were descripted accordingly by reference. 

one more thing is that there are other SNPs in PTPN22 gene which are not considered in this paper: rs1217412, rs1217388, rs1217407, and rs2488458, rs3811021, rs3789605 

  • The SNPs mentioned were added to the manuscript.

You should explain the PTPN22 gene structure and function in a separate heading in the paper 

 In introduction line 60, please add general information about all the pathogenic SNP in PTPN22. How many SNPs are discovered now, how many are coding or non-coding?

  • We improvised the explanation of the PTPN22 gene structure and function in the introduction.

Round 2

Reviewer 1 Report

Accept

Reviewer 2 Report

The revisions and the answers are accepted. It may need some English editing before publication.